# Body Mass Index and Mortality, Recurrence and Readmission after Myocardial Infarction: Systematic Review and Meta-Analysis

**DOI:** 10.3390/jcm11092581

**Published:** 2022-05-05

**Authors:** Lorenzo De Paola, Arnav Mehta, Tiberiu A. Pana, Ben Carter, Roy L. Soiza, Mohannad W. Kafri, John F. Potter, Mamas A. Mamas, Phyo K. Myint

**Affiliations:** 1Aberdeen Cardiovascular & Diabetes Centre, School of Medicine, Medical Sciences & Nutrition, University of Aberdeen, Aberdeen AB24 3FX, UK; lorenzo.depaola.16@abdn.ac.uk (L.D.P.); arnav.mehta.17@abdn.ac.uk (A.M.); tiberiu.pana@abdn.ac.uk (T.A.P.); roy.soiza@nhs.scot (R.L.S.); 2Ageing Clinical and Experimental Research Team, Institute of Applied Health Sciences, University of Aberdeen, Aberdeen AB24 3FX, UK; mwkafri@gmail.com; 3Department of Biostatistics and Health Informatics, Institute of Psychiatry, Psychology and Neuroscience, King’s College London, London WC2R 2LS, UK; ben.carter@kcl.ac.uk; 4Aberdeen Royal Infirmary, NHS Grampian, Aberdeen AB24 3FX, UK; 5Department of Nutrition & Dietetics, Birzeit University, Birzeit P627, Palestine; 6Norwich Medical School, University of East Anglia, Norwich NR4 7TJ, UK; john.potter@uea.ac.uk; 7Keele Cardiovascular Research Group, Keele University, Stoke-on-Trent ST5 5BG, UK; mamasmamas1@yahoo.co.uk

**Keywords:** systematic review, BMI, myocardial infarction, nutrition, mortality, recurrence, hospital readmission

## Abstract

The following study aimed to systematically review and meta-analyse the literature on the relations between markers of nutritional status and long-term mortality, recurrence and all-cause hospital readmission following myocardial infarction (MI). Medline, EMBASE and Web of Science were searched for prospective cohort studies reporting the relationship between anthropometric and biochemical markers of nutritional status and nutritional assessment tools on long-term mortality, recurrence and all-cause hospital readmission in adult patients with an MI. Two reviewers conducted screening, data extraction and critical appraisal independently. Random-effects meta-analysis was performed. Twenty-seven studies were included in the qualitative synthesis and twenty-four in the meta-analysis. All eligible studies analysed BMI as their exposure of interest. Relative to normal weight, mortality was highest in underweight patients (adjusted Hazard Ratio (95% confidence interval): 1.42 (1.24–1.62)) and lower in both overweight (0.85 (0.76–0.94)) and obese patients (0.86 (0.81–0.91)), over a mean follow-up ranging from 6 months to 17 years. No statistically significant associations were identified between different BMI categories for the outcomes of recurrence and hospital readmission. Patients with low BMI carried a significant mortality risk post-MI; however due to the known limitations associated with BMI measurement, further evidence regarding the prognostic utility of other nutritional markers is required.

## 1. Introduction

Obesity and increased adiposity are independent risk factors for incident stroke, myocardial infarction (MI), hypertension, dyslipidaemia, metabolic syndrome, and type 2 diabetes mellitus [1]. The link between obesity and increased cardiovascular disease (CVD) risk in the context of primary prevention has led to the hypothesis that increased fat mass may also be detrimental in secondary prevention settings [2]. Nevertheless, a growing body of evidence suggests that obesity may serve a protective role, conferring a survival advantage amongst populations with established CVD [3]. This phenomenon, termed the “obesity paradox”, has been at the centre of numerous retrospective and prospective epidemiological studies, which have attempted to elucidate the impact of increased adiposity on the clinical outcomes of patients with established CVD [1,3].

At a physiological level, there are various pathways that may contribute to the observed protective effect of obesity. Firstly, obese patients have a greater calorie reserve and therefore a greater ability to endure the catabolic stress associated with acute cardiovascular events [4,5]. Furthermore, although high levels of adiposity are associated with increased levels of pro-inflammatory mediators, which can contribute to the progression of CVDs, adiposity may also contribute to higher levels of anti-inflammatory adipokines, such as interleukin−10 [3]. By inhibiting the release of tumour necrosis factor and other interleukins from macrophages, interleukin−10 may thus contribute to positively modulate deleterious inflammatory processes and confer a survival benefit [6]. This is in contrast to undernourished and underweight patients, who are characterised by lower adiposity, as well as a poorer calorie reserve and a lower degree of muscle mass, which is routinely associated with observed longer lengths of hospital stay, higher rates of complications including infections, pressure sores, falls and overall higher risk of mortality [7].

In this study, we aimed to evaluate the validity of the proposed “obesity paradox” by conducting a systematic review and meta-analysis of the published evidence on the relationship between anthropometric and biochemical markers of nutritional status and clinical outcomes of all-cause mortality, recurrence and hospital readmission, in adult patients following MI.

## 2. Materials and Methods

### 2.1. Protocol and Registration

This review was registered with PROSPERO (https://www.crd.york.ac.uk/prospero, accessed 15 February 2022) (registration number: CRD42021231905). In this paper we focus on the relationship between markers of malnutrition and post-MI outcomes; this is part of a larger search strategy investigating the prospective relationship between nutritional markers and outcomes in CVDs, including both MI and stroke, in populations followed up prospectively.

### 2.2. Eligibility Criteria

The following criteria were applied: (1) prospective cohort studies; (2) patients aged 18 and older with a previous myocardial infarction; (3) assessing one or more of the effects of body mass index (BMI), weight loss, triceps skin fold thickness, creatinine or albumin levels and nutritional screening tools; (4) outcomes including at least one of all-cause mortality, recurrence of cardiovascular event, or hospital readmission. Exclusion criteria included: (1) studies involving patients with unstable angina where distinction between MI and unstable angina was not possible; (2) studies in patients with asymptomatic coronary heart disease; (3) studies in non-English language.

### 2.3. Information Sources

A search of the literature was conducted in duplicate by two independent reviewers (LDP, AM) across the following databases: Medline (Ovid), EMBASE (Ovid) and Web of Science. A combination of MeSH and key/text words were employed to identify the studies and the search strategy modified to suit each database, accordingly, as outlined in Appendix A. Reference lists of included articles were also searched manually to identify further potentially eligible studies.

### 2.4. Study Selection

Searches were conducted on the 11th of January 2021 and later updated on the 6th of June 2021. Results were then transferred to Rayyan review software [8] in order to streamline the study selection process. Based on the eligibility criteria, two reviewers (LDP, AM) independently screened the studies by title, abstract and full text. Consensus between the reviewers was checked within the Rayyan system and discrepancies discussed. In case of disagreement, a decision was reached by consulting a third independent reviewer (TAP).

### 2.5. Data Collection Process

A data extraction form (Appendix A) was designed in order to ensure consistency among reviewers. Data were extracted using the following headings: study characteristics, subject characteristics, study eligibility criteria, baseline CVD, definition of malnutrition, nutrition marker examined, details of intervention and control conditions, outcomes and effect sizes. Following completion of data collection, consensus between the reviewers was checked through discussion and any disagreement was adjudicated by a third reviewer (TAP).

### 2.6. Risk of Bias

Each of the included studies underwent critical appraisal independently by two reviewers (LDP, AM) according to the Scottish Intercollegiate Guidelines Network (SIGN) cohort appraisal checklist [9], assessing the risk of bias according to subject selection, assessment, study confounders and statistical analysis. Reviewers identified age and sex as the main confounders and if a study was observed to not adjust for both, then it was considered to have used inadequate measures to address confounding, and, as such, its overall quality was downgraded. The elements of the SIGN checklist were used to produce a risk of bias summary to be displayed along the forest plots, under the headings of selection, performance, attrition and detection bias and overall study quality. The SIGN checklist and complete critical appraisal are displayed in Appendix A.

### 2.7. Data Synthesis

Only studies that were considered clinically homogeneous in terms of study design, population, outcome, and context were considered for pooling [10].

The odds ratios (ORs), hazard ratios (HRs) and 95% confidence intervals (CI) characterising the relationship between a nutrition marker and the corresponding outcome were extracted, as presented in each study. In the case of studies that only provided the raw outcome without presenting a suitable risk estimate, unadjusted odds ratio and 95% confidence intervals were calculated based on the raw event data, using the formulae displayed in Appendix A. In some records, authors did not report raw data, but presented their results in figures/charts and in such a scenario, the image editing software ImageJ (Version 1.53 e for Windows 10 [11]) was employed to derive data from the graphs in the studies. Forest plots and meta-analyses of the included studies were performed in the Cochrane Collaboration statistical software package, Review Manager (RevMan, Version 5.4.1 for Windows 10 [12]).

Studies that only reported a between group comparison and not individual group summaries were included using a generic inverse variance method and a random-effects model, due to the expected differences between studies. The importance of adjustment for confounding factors was acknowledged, to ensure that the risk estimates extracted would be reflective of the true risk estimate of interest. As it would not have been possible for all of the studies to adjust for the same set of confounders, it was deemed necessary to establish a set of minimum common variables to adjust for. Age and sex were thus considered the minimum set of common adjusting variables required to deem estimates as ‘adjusted’.

Heterogeneity was assessed by reviewing study characteristics. Where substantial variation in study design and baseline population was observed, data were analysed narratively. Where no evidence of significant study design or population heterogeneity was encountered, a pooled meta-analysis was performed. Statistical heterogeneity was assessed using the I-squared (I^2^) statistic, where an I^2^ of 0%, 25%, 50%, and 75% corresponded to a no, low, moderate, and high level of heterogeneity, respectively [13].

Publication bias was assessed in *RevMan* using a funnel plot of the primary outcome of all-cause mortality.

### 2.8. Additional Analyses

Sensitivity analyses were undertaken by examining both statistical heterogeneity as expressed by the I^2^, and heterogeneity in study design and population characteristics. Based on these factors, a set of eligibility criteria were developed, as suggested in the *Cochrane Handbook of Systematic Reviews* [10]. These criteria included: mean follow-up time, population mean age, total number of participants and study design (registry analysis versus single-centre study). These criteria allowed the identification of studies that contributed to a significant proportion of heterogeneity. Sensitivity analyses were subsequently performed by excluding these studies. Sensitivity analyses were only performed for the main analyses assessing adjusted HR, as these likely represent the most clinically significant results.

## 3. Results

### 3.1. Study Selection

The study selection process is summarised in a PRISMA flow diagram (Figure 1). Following duplicate removal, a total of 5096 studies were identified from the search. After title, abstract and full-text analysis screening, 27 articles were eligible to be included in this study.

### 3.2. Study Characteristics

The characteristics of all included studies are summarised in Table 1. Respective risk estimates are displayed in Appendix A. Of the 27 studies that met the eligibility criteria, 26 examined the outcome of all-cause mortality following MI, 9 the odds of MI recurrence and 4 the odds of readmission at follow-up. Despite the search strategy including other markers of malnutrition, all eligible studies only used BMI as their predictor. A total of 308,430 participants (132,759 women and 175,671 men) were included. The mean follow-up time across the studies ranged from 6 months to 17 years. Thirteen studies were conducted in the USA [14,15,16,17,18,19,20,21,22,23,24,25,26], three each in Germany [27,28,29] and Japan [30,31,32], two in South Korea [33,34], one each in Australia [35], Croatia [36], China [37], Denmark [38], France [39] and Israel [40].

### 3.3. Critical Appraisal

Overall, the studies were found to be of moderate to high methodological quality and only three studies [17,27,36] were considered to be of low quality, as defined by the SIGN checklist [9].

Common strengths across the studies included: addressing an appropriate and clearly focused question (*n* = 27; 100%), comparing participants that were similar in all respects except for their exposure status (*n* = 27; 100%), clearly defining the number of participants in each group (*n* = 27; 100%), having appropriately defined outcome measures (*n* = 27; 100%) and reliably assessing the exposure (*n* = 24; 89%). Only two of the included studies [29,33] showed significant patient attrition, i.e., 20% or greater. Additionally, the majority of studies provided confidence intervals (*n* = 20; 74%) and found a clear association between exposure and outcome (*n* = 23; 85%) The follow-up was of suitably adequate duration in all of the studies (minimum 6 months). A number of relevant limitations were notable across the selected articles. Out of the two major confounding variables identified (age and sex), two studies [35,38] (7%) adjusted for only one variable, six studies [17,27,30,36,37,40] (22%) did not adjust for either, and one study [39] adjusted for variables other than age and sex. The majority of studies (*n* = 22; 81%) did not blind assessors to the exposure status and only two studies [18,37] assessed the exposure more than once.

No significant publication bias was identified through interpretation of a funnel plot of the primary outcome of all-cause mortality (Appendix A).

### 3.4. Long-Term Mortality following MI in Overweight Patients Compared to Normal Weight

Twenty-two studies assessing the risk of mortality following MI in overweight patients were eligible for inclusion in the meta-analysis of long term mortality. The exposure was BMI 25.0–29.9 kg/m^2^, while the reference group consisted in BMI 18.5–24.9 kg/m^2^. Studies were separately pooled based on their respective risk estimate and meta-analysed. Fourteen studies reported their effect as unadjusted ORs and eight as HRs. Six studies provided both unadjusted and adjusted HRs, while two only provided adjusted HRs.

The pooled adjusted HR meta-analysis following inclusion of eight studies [14,15,18,21,23,24,28,32] displayed a 15% reduction in the risk of all-cause mortality in overweight individuals compared to normal weight (aHR: 0.85 0.76–0.94, *p* = 0.002), as shown in the forest plot in Figure 2. Unadjusted HR meta-analysis after the inclusion of six studies [14,15,21,23,24,28] displayed a similar effect with a HR of 0.67 (0.59–0.77, *p* <0.0001) (Appendix A). Pooled analysis of the fourteen studies [17,19,20,22,27,29,30,31,34,35,36,37,38,40] reporting unadjusted ORs displayed lower odds of mortality in overweight relative to normal weight individuals (OR: 0.72, 0.64–0.81, *p* < 0.0001) (Appendix A).

Statistical heterogeneity was found to be moderate to high, with an I^2^ of 63% in the unadjusted OR subgroup, 97% in the unadjusted HR and 86% in the adjusted HR pool, which therefore suggested a substantial degree of variation across the studies. Sensitivity analysis of the adjusted HR meta-analysis was performed by excluding three studies. O’Brien et al. [23] and Bucholz et al. [15] were excluded due to both including registry data with a very large sample size compared to the rest of the studies. As shown in Appendix A, Fukouka et al. [32] split its population by age. It was therefore deemed appropriate to exclude data from the younger age group, due to their significantly lower mean age than the rest of the included population (56 versus 67 years), and thus was likely a substantial contributor of heterogeneity. Exclusion of the three studies resulted in the I^2^ being decreased to 6%, indicating a low level of heterogeneity. No substantial change in effect was observed following sensitivity analysis (aHR: 0.81, 0.72–0.92, *p* = 0.0007) (Appendix A).

### 3.5. Long-Term Mortality Following MI in Obese and Morbidly Obese Patients Compared to Normal Weight

Results from each of the studies examining the association between obesity and long-term mortality following an MI are summarised in Appendix A. Across the included studies, obesity was defined as a BMI >30 kg/m^2^ and morbid obesity as a BMI >35 kg/m^2^, while normal weight was the reference category, with a BMI range of 18.5–24.9 kg/m^2^. A total of nineteen studies were found to be eligible for inclusion in the meta-analysis and, of these, twelve reported ORs and seven HRs. Of the studies reporting HRs, adjusted estimates for both age and sex were available for seven, while only unadjusted risk estimates were compared for the studies using ORs. Studies were pooled based on their estimate and their effect sizes compared.

Meta-analysis of adjusted HR estimates following the inclusion of seven studies [14,15,18,21,23,24,28] displayed a lower risk of all-cause mortality in obese relative to normal weight individuals (aHR: 0.86, 0.81–0.91 *p* < 0.0001) (Figure 2). On the other hand, pooled analysis of the four studies [14,15,23,24] reporting adjusted HR estimates only for morbidly obese patients did not show any statistically significant difference in mortality, relative to normal weight (HR: 0.89, 0.78–1.01, *p* = 0.08) (Figure 2). The meta-analysis of the studies [14,15,21,23,28] reporting unadjusted HRs showed a similar effect size and direction, with a risk of 0.64 (0.56–0.73, *p* < 0.00001) for the unadjusted estimates (Appendix A). Similarly, across the twelve studies [17,19,20,22,27,29,31,35,36,37,38,40] using unadjusted ORs, the odds of mortality following MI were reduced by 38% in obese compared to normal weight individuals (OR: 0.62, 0.50–0.77, *p* < 0.0001) (Appendix A).

Statistical heterogeneity, as assessed by I^2^, was found to be high with 79% in the studies reporting unadjusted ORs, 98% in studies with unadjusted HRs and 88% in those reporting adjusted HRs. Sensitivity analysis was performed for studies with adjusted HRs. This resulted in two studies being excluded [15,23], due to both reporting registry data with a considerably large number of participants. Following their removal, heterogeneity decreased to 69%, suggesting a moderate level of variation between the studies. Sensitivity analysis caused slight variation in the overall effect direction and size (HR: 0.79, 063–1.00, *p* = 0.05) (Appendix A).

### 3.6. Long-Term Mortality following MI in Underweight Patients Compared to Normal Weight

Eight studies assessing mortality following MI in underweight (BMI <18.5 kg/m^2^) versus normal weight patients, (BMI 18.5–24.9 kg/m^2^) were meta-analysed. Of these, three provided unadjusted ORs and five HRs. Three studies provided both unadjusted and adjusted HRs, while two only provided adjusted HRs. Studies were grouped based on their estimate and their effect sizes compared.

Overall, underweight patients appeared to have a worse outcome compared to normal weight. The pooled adjusted HR meta-analysis following inclusion of five studies [16,18,23,32,34] displayed a 42% greater risk of all-cause mortality following MI in underweight compared to normal weight patients (aHR: 1.42, 1.25–1.62, *p* < 0.0001) (Figure 2). Across the unadjusted HR group [16,23,34], the risk of mortality was 1.96 (1.63–2.36, *p* < 0.0001) (Appendix A). Pooled analysis of studies reporting ORs [31,38,40] showed increased odds of mortality at follow-up for underweight participants (OR: 2.48, 1.77–3.47, *p* < 0.0001) (Appendix A).

Heterogeneity indicated by I^2^ was found to be 14% for the unadjusted OR group and 93% and 79% for unadjusted and adjusted HRs, respectively. Sensitivity analysis of the adjusted HR meta-analysis was performed. Three studies were removed [16,23,32], which caused heterogeneity to decrease to 0%. Following sensitivity analysis, the overall effect was HR 1.61 (1.30–1.98, *p* < 0.0001) (Appendix A).

### 3.7. Studies Not Included in Meta-Analysis

Three studies could not be included in the meta-analysis [26,33,39].

The study by Wu et al. [26] explored the association between high BMI and all-cause mortality following MI. In this study, the risk of mortality of obese individuals was compared to a category incorporating both normal weight and overweight individuals. Since the control group of the study was not comparable to that of all of the other included records, the HR provided could not be pooled. In their unadjusted analysis, the authors observed a lower risk of mortality in obese patients (HR: 0.82, 0.70–0.95, *p* = 0.008), which was, however, rendered non-significant following adjustment for age (HR: 0.91, 0.78–1.06, *p* = 0.206).

Zeller et al. [39] assessed the combined effect of waist circumference and BMI on the risk of long-term mortality following MI. No statistically significant differences in mortality were observed between waist circumference tertiles. In men, lower mortality rates were found with higher BMI tertiles, while no significant difference was observed in women. When BMI was analysed as a continuous variable, higher BMI was found to have a protective effect, with a 5% reduction in the risk of mortality for each unit increase in BMI.

Kang et al. [33] employed different BMI ranges. The findings showed that higher BMI was associated with lower post-event mortality, while underweight patients were associated with higher mortality. Higher BMI was associated with improved clinical outcomes at one year follow-up, with a mortality rate of 15.4% in underweight (BMI < 18.5 kg/m^2^), 3.3% in normal weight (BMI 18.5–23.0 kg/m^2^), 2.6% in overweight (BMI 23–27.5 kg/m^2^), and 1.1% in obese (BMI ≥ 27.5 kg/m^2^) patients.

### 3.8. The Association between BMI and MI Recurrence on Hospital Readmission

Data regarding the risk of MI recurrence in overweight patients, BMI 25–30 kg/m^2^, relative to normal weight, BMI 18.5–24.9 kg/m^2^, were available for ten studies. One study [21] provided both adjusted and unadjusted HRs, while the remaining seven only provided the raw number of MI recurrence at follow-up. As the only study reporting HR, Nigam et al. [21] was excluded from the meta-analysis. However, its data are summarised in Appendix A. The calculated odds ratios from the ten included studies [17,18,19,25,27,30,34,35,36,37] were meta-analysed, as displayed in the forest plot in Figure 3. The overall effect from the pooled studies did not show a statistically significant change in MI recurrence in overweight compared to normal weight individuals (OR: 0.88, 0.79–1.01, *p* = 0.07). Heterogeneity was found to be 0%, and thus sensitivity analysis was not performed.

Eight studies [17,18,19,25,27,35,36,37] reporting MI recurrence in obese (BMI > 30 kg/m^2^) versus normal weight patients were identified. Meta-analysis was performed, as summarised in Figure 3. The overall effect suggested no statistically significant difference in the odds of recurrence of MI in obese relative to normal weight patients (OR: 1.10, 0.90–1.33, *p* = 0.35) The I^2^ was calculated to be 21%, indicating low heterogeneity.

Only two [18,34] of the studies provided sufficient data to examine the risk of MI recurrence in underweight patients, BMI < 18.5 kg/m^2^. These studies only provided raw number of events, which were used to calculate unadjusted ORs and meta-analysed, as shown in Figure 3. No statistically significant difference in odds of recurrence of MI in underweight compared to normal weight patients was found (OR: 1.15; 0.84–1.58, *p* = 0.37, I^2^ = 0%).

Three studies [30,36,37] investigated readmission in overweight compared to normal weight patients diagnosed with an MI. The overall effect size was found to not be statistically significant (OR: 0.72, 0.31–1.70, *p* = 0.46), as shown in Figure 4. Individual study results with risk estimates assessing the risk of readmission in obese compared to normal weight in patients diagnosed with an MI are presented in Appendix A. Three studies investigated readmission in obese compared to normal weight patients. Two studies [36,37] were included for meta-analysis. No statistically significant differences between the two groups was identified (OR: 0.79, 0.49–1.28, *p* = 0.33), as displayed in Figure 4. Only one study [23] assessed the readmission in underweight relative to normal weight, finding a higher risk in the former (HR: 1.16, 1.08–1.25)

## 4. Discussion

In this study we aimed to conduct a systematic review and meta-analysis on the relationship between anthropometric and biochemical markers of nutritional status and important outcomes in patients with MI. In total, 27 prospective studies including 308,430 patients were found to be eligible for inclusion. All studies, however, only assessed the associations between BMI and selected outcomes over a mean follow-up ranging from 6 months to 17 years.

We found that both overweight and obese patients had ~15% lower risk of post-MI mortality, compared to normal weight individuals. On the other hand, morbidly obese patients did not display any statistically significant differences in mortality outcome compared to their normal weight counterparts, albeit this could be due to smaller sample size. Strikingly, underweight patients had a 42% higher risk of long-term post-MI mortality. Regarding the secondary outcomes of MI recurrence and hospital readmission, our analysis did not highlight any association between BMI and post-MI outcomes.

There are several mechanisms which may explain the observed associations. Firstly, it has been suggested that the greater caloric reserve at disposal of overweight and obese patients could be advantageous in the recovery from an acute cardiovascular event and confer a protective effect in the long-term after MI [5]. It should, however, be noted that that the sole use of BMI as an indicator of under or overnutrition makes the interpretation of such relationship challenging, since, unlike waist circumference, BMI does not represent an effective tool to evaluate adiposity and, more specifically, central adiposity, which has been shown to be associated with a substantial cardiometabolic risk [41]. Moreover, BMI has been shown to not be sensitive enough in the evaluation of small but clinically meaningful weight loss, as significant BMI changes within the normal reference range would not be highlighted as meaningful, despite the fact that they may still be associated with increased morbidity and mortality [42]. To this regard it has been previously suggested that a BMI reference range of 20–25 kg/m^2^ might indeed not be appropriate for older individuals [42]. Indeed, Landi et al. [43] have shown that among older people (81.2 ± 7.3 years) living in the community, a BMI > 27 kg/m^2^ did not represent a risk factor for mortality, whereas a BMI < 22 kg/m^2^ was strongly associated with a greater risk of mortality.

Our study further highlights the negative prognostic impact of low BMI in cardiovascular patients, as we report a majorly elevated risk of long-term all-cause mortality following MI associated with lower than normal BMI (42% increased relative risk). It should, however, be noted almost all of the included studies assessed clinical outcomes in older patients, with a mean age >65 years, and thus it could be argued that such a striking result may be more strongly related to sarcopenia, rather than a low level of adiposity per se. Indeed, sarcopenia is associated with frailty in older adults and often leads to reduced mobility, exercise capacity, worse cardiovascular fitness and decreased overall survival [44]. The exclusive use of BMI as a nutritional screening tool in the selected studies, however, once again limits the interpretations of our results and prevents us from fully evaluating the true relationship between proportions of fat mass and fat-free mass and their respective effects on long-term survival. Overall, the lack of studies on other forms of nutritional assessments highlights the limitations in the current available literature and the still limited use of more detailed nutritional screening in clinical practice. Nevertheless, our finding does emphasise the importance of routine nutritional screening in the acute setting and identification of cardiovascular patients who are clinically underweight, so that prompt and targeted nutritional management can be conducted, which could potentially save lives.

Although a subgroup analysis by age could not be performed, the study by Fukuoka et al. [32] suggested that age might indeed play a critical role in the association between BMI and cardiovascular mortality. Interestingly, in this study, a high BMI was only associated with lower mortality in patients over 70 years of age, whereas those under 70 had a risk of mortality at follow-up that was three times higher than that of normal weight participants. It is, therefore, plausible to assume that in older patients a higher BMI indicates preserved lean mass and an overall lower degree of frailty, rather than a higher level of adiposity per se. Only a few studies [15,23,24,40,45] separated obese patients into further subgroups (e.g., Class I, II and III). Nevertheless, it should be noted that in these studies patients in the upper threshold of the obese class appeared to suffer from a higher risk of long-term mortality, compared to those at the lower end of obesity. This would, therefore, suggest that the association between BMI and cardiovascular mortality may follow a J-shaped pattern, rather than U-shaped, as previously referred to in the literature [46,47].

The results of our study are consistent with previous similar systematic reviews and meta-analyses [48,49]. While previous systematic reviews evaluated the impact of BMI in patients with established coronary artery disease, our review only focused solely on post-MI individuals. Despite our study including a more strictly defined population, our analysis showed relatively similar risks of long-term mortality across the different BMI categories. It is striking to note that in a similar way the findings of Wang et al. [48] also highlight the existence of a J-shaped trend, with more severely obese patients being at higher risk, compared to those with less severe obesity.

No significant differences in MI recurrence and hospital readmission were found in this evidence synthesis. This could be due to the relatively smaller number of studies and further research required for these outcomes. A growing evidence base suggests that a lower than normal BMI might indeed constitute a significant risk factor for the development and progression of CVDs. The cross-sectional study conducted by Park and colleague [50] included over 400,000 participants and showed that patients with a BMI below 18.5 kg/m^2^ had a higher incidence of cardiovascular diseases than patients in the normal BMI range, which remained statistically significant even following adjustments for confounders. Few studies have been conducted to attempt to explain the mechanism behind such phenomenon; however, it has been suggested that clinical risk factors associated with being underweight, such as ageing, sarcopenia and nutrient deficiency could be responsible. One other possible explanation that has been recently proposed in a major Japanese cohort study [51], is the concept of “metabolically obese underweight”. This refers to patients who are phenotypically underweight, but display proportions of visceral fat, as well as metabolic abnormalities, such as dyslipidaemia and insulin resistance, that would be typical of phenotypically obese patients.

Our systematic review and meta-analysis has several strengths. Firstly, a comprehensive and thorough search strategy was implemented. The quality control and double reviewer process were employed throughout the process to ensure the robust methodology. The presence of publication bias was also assessed through a funnel plot and found to be absent. The studies included in the meta-analysis used the same comparison group, normal BMI category, which ensured that the risks estimates abstracted for each predictor shared the same comparison group characteristics. Moreover, although we excluded non-English language studies, papers included originated from different world regions, and thus our results are likely to be generalisable, especially given that the effect of BMI on outcomes is unlikely to differ between different races or languages.

We also acknowledge some limitations. As a systematic review we are limited by the quality of original studies and that has led to considerable heterogeneity of the results. The exclusion of studies following sensitivity analysis, however, did not lead to significant changes in direction or size of the effect. It is plausible that such heterogeneity might be due to the large number of events and considerable sample size, suggesting that our meta-analysis had sufficient statistical power to recognise even minor differences across the studies for the long-term mortality outcome. As previously described by Romero-Corral et al. [49], such large variability may, therefore, represent a strength rather than a limitation of this review. Additionally, included studies relied on single BMI measurements captured mostly at hospital admission, which make the data prone to reverse causality bias, as patients admitted to hospital because of their MI, and who may have started losing weight because of an underlying sickness, are included. It is, therefore, important to acknowledge that the lack of information about weight change and its nature, whether this is intentional or unintentional, as well as the selection of patients in these studies, which are mostly from hospital registries or claims data, and thus contributing to selection bias, may have been partly responsible for the measured ‘obesity paradox’.

## 5. Conclusions

Patients with low BMI carry a significant mortality risk post-MI and there is a clear prospective association between a higher BMI and a lower risk of long-term mortality following MI. Our findings emphasise the limitations in the current available literature, highlighting the lack of studies assessing alternative anthropometric and biochemical assessment techniques. We thus recommend future research focusing on the prognostic utility of other markers of nutritional status in MI, to better identify those who are at increased risk of poor clinical outcomes in patients with MI.

## Figures and Tables

**Figure 1 jcm-11-02581-f001:**
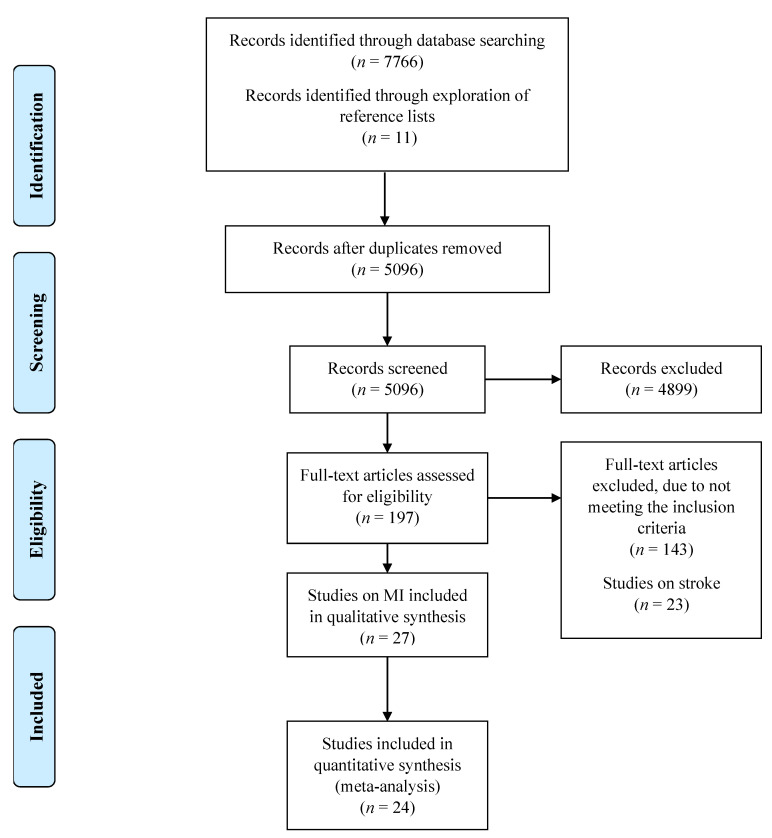
PRISMA flow diagram displaying literature search protocol. Adapted from PRISMA 2009 Flow Diagram.

**Figure 2 jcm-11-02581-f002:**
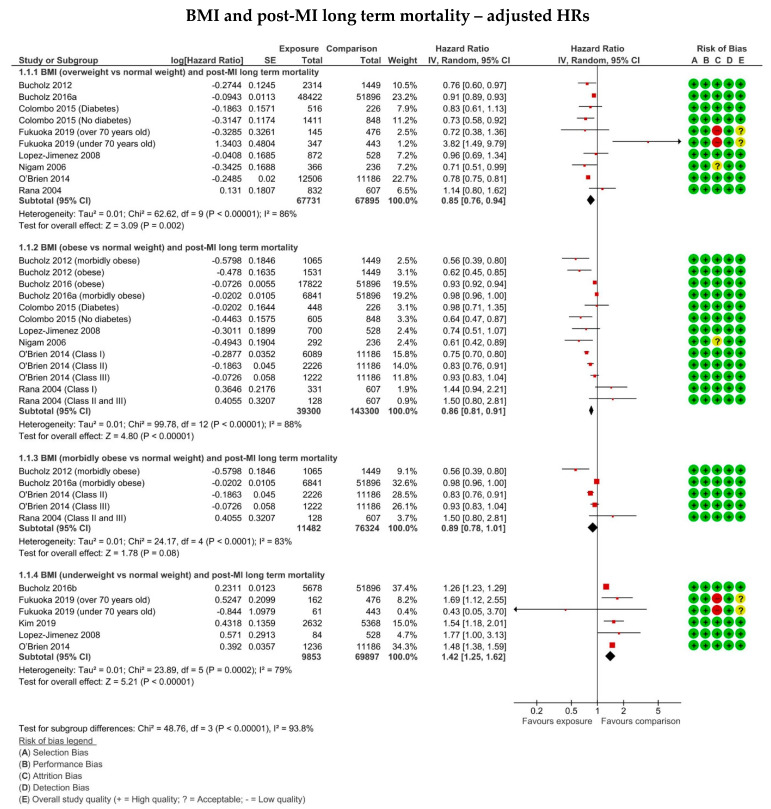
Forest plots displaying the risk of long-term mortality following MI in overweight (BMI 25–29.9 kg/m^2^), obese (BMI ≥ 30 kg/m^2^), morbidly obese (BMI ≥35 kg/m^2^) and underweight (BMI < 18.5 kg/m^2^) patients compared to normal weight (BMI 18.5–24.9 kg/m^2^) for the studies using adjusted hazard ratio. The risk of bias summary produced from the critical appraisal according to the SIGN cohort appraisal checklist is also displayed for each included study.

**Figure 3 jcm-11-02581-f003:**
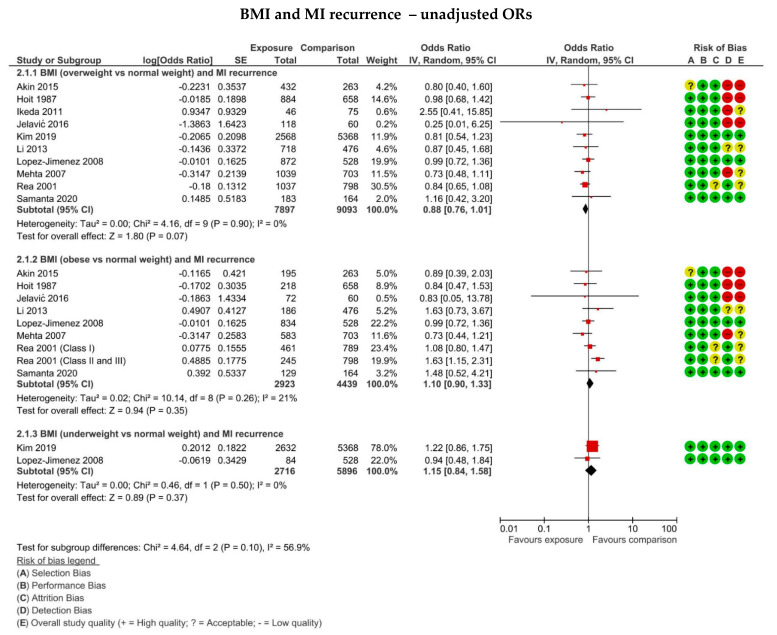
Forest plots displaying the risk of MI recurrence in overweight (BMI 25–29.9 kg/m^2^), obese (BMI ≥ 30 kg/m^2^), morbidly obese (BMI ≥ 35 kg/m^2^) and underweight (BMI < 18.5 kg/m^2^) patients compared to normal weight (BMI 18.5–24.9 kg/m^2^) for the studies using unadjusted odds ratio. The risk of bias summary produced from the critical appraisal according to the SIGN cohort appraisal checklist is also displayed for each included study.

**Figure 4 jcm-11-02581-f004:**
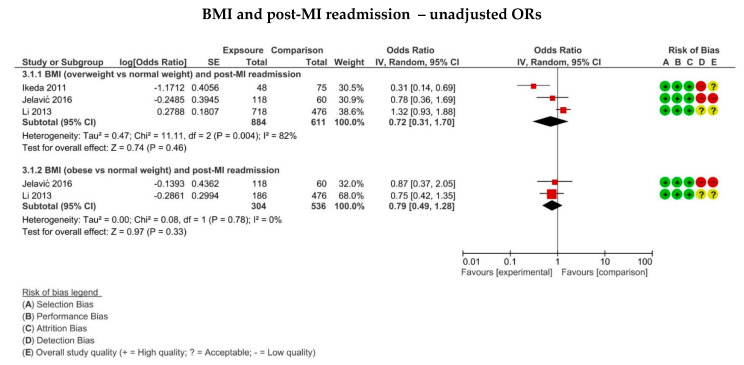
Forest plots displaying the risk of post-MI hospital readmission in overweight (BMI 25–29.9 kg/m^2^) and obese (BMI ≥ 30 kg/m^2^) patients compared to normal weight (BMI 18.5–24.9 kg/m^2^) for the studies using unadjusted odds ratio. The risk of bias summary produced from the critical appraisal according to the SIGN cohort appraisal checklist is also displayed for each included study.

**Table 1 jcm-11-02581-t001:** Characteristics of included studies using anthropometric nutrition markers included in the systematic review. Not all studies were included in the meta-analysis.

Study	Follow-up (months)	Females/ Males	Country	Exposure	Comparison	Outcome Assessed
**Akin** **2015** [27]	12	205/ 685	Germany	BMI > 30, 25–30 kg/m^2^	BMI ≤ 24.9 kg/m^2^	Mortality, recurrence
**Aronson** **2010** [40]	26	459/ 1698	Israel	BMI ≥ 35, 30–34.9, 25–29.9, < 18.5 kg/m2	BMI 18.5–24.9 kg/m^2^	Mortality
**Bucholz** **2012** [14]	12	2076/ 4283	USA	BMI ≥ 35, 30–34.9, 25–29.9 kg/m^2^	BMI 18.5–24.9 kg/m^2^	Mortality
**Bucholz** **2016 a [15]**	204	57,921/ 67,060	USA	BMI ≥ 35, 30–34.9, 25–29.9 kg/m^2^	BMI 18.5–24.9 kg/m^2^	Mortality
**Bucholz** **2016 b** [16]	204	29,258/ 28316	USA	BMI < 18.5 kg/m^2^	BMI 18.5–24.9 kg/m^2^	Mortality
**Colombo 2015** [28] **(Diabetes)**	120	337/ 853	Germany	BMI ≥ 30, 25–29.9 kg/m^2^	BMI 18.5–24.9 kg/m^2^	Mortality
**Colombo 2015** [28] **(No diabetes)**	120	630/ 2234	Germany	BMI > 30, 25–29.9 kg/m^2^	BMI 18.5–24.9 kg/m^2^	Mortality
**Fukuoka** **2019** [32]	12	454/ 1666	Japan	BMI ≥ 25, <20 kg/m^2^	BMI 20–24.9 kg/m^2^	Mortality
**Hoit** **1987** [17]	12	433/ 1327	USA	BMI > 30, BMI 25–30 kg/m^2^	BMI < 25 kg/m^2^	Mortality, recurrence
**Ikeda** **2011** [30]	60	21/ 100	Japan	BMI ≥ 25 kg/m^2^	BMI < 25 kg/m^2^	Mortality, recurrence, readmission
**Jelavić** **2016** [36]	12	73/ 177	Croatia	BMI ≥ 30, 25–29.9 kg/m^2^	BMI < 25 kg/m^2^	Mortality, recurrence, readmission
**Kang** **2010** [33]	12	928/ 2896	South Korea	BMI ≥ 27.5, 23–27.5, <18.5 kg/m2	BMI 18.5–23 kg/m2	Mortality
**Kim** **2019** [34]	12	2547/ 8021	South Korea	BMI ≥ 26, <22 kg/m^2^	BMI 22–26 kg/m^2^	Mortality, recurrence
**Kragelund** **2005** [38]	120	2172/ 4502	Denmark	BMI ≥ 30, 25–29.9, <19 kg/m^2^	BMI 20–25 kg/m^2^	Mortality
**Li** **2013** [37]	12	1380 (total)	China	BMI ≥ 28.0, 25–28 kg/m^2^	BMI 18.5–24 kg/m^2^	Mortality, recurrence, readmission
**Lopez-Jimenez 2008** [18]	6.2	1022/ 1296	USA	BMI ≥ 30, 25–29.9, < 19 kg/m^2^	BMI 20–25 kg/m^2^	Mortality, recurrence
**Mehta** **2007** [19]	12	606/ 1719	USA	BM ≥ 30 kg/m^2^	BMI < 25 kg/m^2^	Mortality
**Neeland** **2017** [20]	36	7397/ 12102	USA	BMI ≥ 40, 35–39.9, 30–34.9, 25–29.9 kg/m^2^	BMI 18.5–24.9 kg/m^2^	Mortality
**Nigam** **2006** [21]	12	278/ 616	USA	BMI ≥ 30 kg/m^2^	BMI < 25 kg/m^2^	Mortality, recurrence
**Nikolsky** **2006** [22]	12	542/ 1493	USA	BMI ≥ 30, 25–30 kg/m^2^	BMI < 25 kg/m^2^	Mortality
**O’Brien** **2014** [23]	36	16,351/ 18,114	USA	BMI ≥ 40, 35–39.9, 30–34.9, 25–29.9, < 18.5 kg/m^2^	BMI 18.5–24.9 kg/m^2^	Mortality, readmission
**Rana** **2004** [24]	45	1317/ 581	USA	BMI ≥ 30, 25–29.9 kg/m^2^	BMI 20–25 kg/m^2^	Mortality
**Rea** **2001** [25]	36	968/ 1573	USA	BMI ≥ 30 kg/m^2^	BMI < 25 kg/m^2^	Recurrence
**Samanta** **2020** [35]	12	77/ 399	Australia	BMI > 30, 25–29.99 kg/m^2^	BMI 18.5–24.9 kg/m^2^	Mortality, recurrence
**Wienbergen 2008** [29]	14	3137/ 7397	Germany	BMI ≥ 30, 25–29.9 kg/m^2^	BMI 18.5–24.9 kg/m^2^	Mortality
**Wu** **2010** [26]	16	1885/ 4675	USA	BMI ≥ 30, 25–29.9 kg/m^2^	BMI < 25 kg/m^2^	Mortality
**Yokoyama** **2019** [31]	32.4	112/ 405	Japan	BMI ≥ 26.0, ≥24- < 26, <21.9 kg/m^2^	BMI ≥ 21.9- <24 kg/m^2^	Mortality
**Zeller** **2008** [39]	12	593/ 1636	France	BMI ≥ 30, 25–29.9 kg/m^2^	BMI < 25 kg/m^2^	Mortality

## Data Availability

All data are incorporated into the article and its Appendix A.

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
