# Peer review of "Body Mass Index and Mortality, Recurrence and Readmission after Myocardial Infarction: Systematic Review and Meta-Analysis"

_jcm, 2022, doi:10.3390/jcm11092581_

Round 1
Reviewer 1 Report
I consider that the idea of this study, to identify the relations between markers of nutritional status and long-term mortality, recurrence, and all-cause hospital readmission following myocardial infarction, is very interesting and with important clinical implications. The authors made a systematic review and observed that patients with low body mass index have a significant mortality risk after myocardial infarction. The article is well-structured and the results are presented in an appropriate manner. The statistical analysis is also well done.
Author Response
Thank you very much for taking the time to review our manuscript and for providing your expert opinion on our work.
Reviewer 2 Report
This is an interesting and thoughtful contribution.
What was the hypothesis that was being tested in this study? Please include at end of Introduction.
Author Response
Thank you very much for taking the time to review our manuscript and for providing your expert opinion on our work.
We appreciated your suggestion regarding the introduction and we have revised the manuscript accodingly. Please see the attached file containing the revised manuscript with tracked changes.

Reviewer 3 Report
I’ve read with attention the paper of De Paola et al. that is potentially of interest. The background and aim of the study have been clearly defined. The methodology applied is overall correct, the results are reliable and adequately discussed. It could be interesting to split the large category of overweight in two (25-27.5 and 27.5-30), at least in a figure showing the trend between BMI and the considered outcomes. The conclusion should be more short and focused and not include part of the discussion section.
Author Response
Thank you very much for taking the time to review our manuscript and for providing your expert opinion on our work.
Unfortunately due to the nature of the systematic review we have conducted, we are limited in our analyses by the data available in the included articles. The included articles used the WHO definition of overweight (i.e. BMI between 25 and 29.9) and only provided estimates for this category as a whole and did not explore BMI as a continous variable. As such it would not be possible for us to further split the overweight category into two separate subgroups and analyse accodingly.
We appreciated your suggestion regarding the conclusion and we have revised the manuscript accodingly. Please see the attached file which contains the revised manuscript with tracked changes.
